# Effect of Rituximab Compared with Natalizumab and Fingolimod in Patients with Relapsing–Remitting Multiple Sclerosis: A Cohort Study

**DOI:** 10.3390/jcm11133584

**Published:** 2022-06-22

**Authors:** Martha Rocio Hernández-Preciado, Jazmin Marquez-Pedroza, Nayeli Alejandra Sánchez-Rosales, José de Jesús García-Rivera, Antonio Kobayashi-Gutiérrez, Blanca Miriam Torres-Mendoza, Efraín Chavarría-Avila, Raúl Alejandro Montaño-Serrano, Fernando Cortes-Enriquez, Mario Alberto Mireles-Ramírez

**Affiliations:** 1High Specialty Medical Unit, Western National Medical Center of the Mexican Institute of Social Security, Guadalajara 44340, Mexico; mrociohp@hotmail.com (M.R.H.-P.); jaz180688@gmail.com (J.M.-P.); naye_ale@hotmail.com (N.A.S.-R.); garciar10@hotmail.com (J.d.J.G.-R.); drkoba@hotmail.com (A.K.-G.); 2Neurosciences Division, Western Biomedical Research Center (IMSS), Guadalajara 44340, Mexico; bltorres1@hotmail.com; 3Department of Philosophical and Methodological Disciplines, University Health Sciences Center, University of Guadalajara, Guadalajara 44340, Mexico; efrain.chavarria@academicos.udg.mx; 4Department of Physiology, University of Guadalajara, Guadalajara 44340, Mexico; dr.alexmontano@gmail.com; 5Department of Neurology, Hospital General Regional No 45 of the Mexican Institute of Social Security, Guadalajara 44910, Mexico; fercorts08@gmail.com

**Keywords:** multiple sclerosis, relapsing–remitting, rituximab

## Abstract

The objective of this study was to evaluate the clinical files of patients with RRMS who started rituximab (RTX) compared with a second-line treatment (natalizumab (NTZ) or fingolimod (FTY)). This was a historical cohort study. We compared the effect according to the Expanded Disability Status Scale (EDSS) and the number of relapses in RRMS patients receiving these treatments after a mean period of 12 months. We found a statistically significant difference (*p* < 0.001) when comparing the EDSS scores and the annual relapse rates of patients receiving RTX with those receiving NTZ or FTY. This study is essential for our clinical practice, since patients with limited treatment options represent a challenge with regard to the management of their medical care. However, clinical trials and prospective studies with long follow-up periods are necessary to provide sufficient evidence on the efficacy of RTX and thus include this treatment in the therapeutic profile of patients with MS.

## 1. Introduction

Multiple sclerosis (MS) is a chronic, autoimmune, demyelinating, and neurodegenerative disease that exclusively affects the central nervous system (CNS) [1]. MS affects around 2.5 million people worldwide [2] and is the leading cause of non-traumatic neurological disability in young patients [3]. The first phase of the relapsing–remitting multiple sclerosis (RRMS) phenotype is characterized by clinical relapses, which are usually followed by functional recovery [4].

Eleven disease-modifying therapies (DMTs) are used to prevent the accumulation of CNS lesions and, therefore, transient and/or permanent neurological deficits, according to the evolution of the MS disease. However, there is still no proven curative therapy for MS [1,4,5].

Natalizumab (NTZ) is a highly effective DMT according to the outcomes of two-year, phase III studies. These studies, which were carried out in patients with relapsing MS, demonstrated that NTZ could significantly reduce the annualized relapse rate, the risk of confirmed disability deterioration over two years, and the accumulation of new brain MRI lesions [6,7]. The efficacy and safety of fingolimod (FTY) were shown in two-year, phase III clinical trials, named the FREEDOMS I trial and its extension, FREEDOMS II. These studies showed that FTY reduced the risk of progression of disability and the number of inflammatory lesions and improved several measures of cranial magnetic resonance and cerebral atrophy [8,9].

Although the efficacy of NTZ and FTY has been shown previously, the appearance of progressive multifocal leukoencephalopathy and the risk of opportunistic infections related to lymphopenia, macular edema, and rare cardiological anomalies are factors in the discontinuation of these treatments [10,11].

Rituximab (RTX) is a monoclonal antibody that lyses CD20 lymphocytes and causes the prolonged depletion of circulating B lymphocytes [12]. RTX treatment is effective in diseases in which autoantibodies are an important part of the pathogenesis, such as MS [13]. RTX has been found to be a highly effective alternative after the discontinuation of second-line treatments such as NTZ and FTY [14].

Studies have shown that RTX is able to reduce disease activity, inflammatory brain lesions, and clinical relapses in RRMS [15]. Studies such as OLYMPUS conducted in primary progressive multiple sclerosis did not meet their primary efficacy endpoint. However, from baseline to week 96, the patients showed a lower increase in T2 lesion volume, and subgroup analysis showed the delayed progression of disability in patients under 51 years. [16]. Additionally, this substance has been shown to have better effectiveness over a long period than FTY and dimethyl fumarate [17].

The advantages of using RTX in patients with RRMS include its reasonable cost; the existing knowledge of its safety profile (due to the time for which the medication has been on the market for other pathologies); and its availability in health institutions, which is worth mentioning because the treatment options for this disease are limited in the Mexican public healthcare system [18,19].

Even though, in clinical practice, RTX is considered an off-label drug, in our center, it is administered when the patient has a score on the Expanded Disability Status Scale (EDSS) equal to or greater than 5.5 and when there has been therapeutic failure with other DMTs, a high lesion burden, the progression of disability, multiple relapses, contraindications, or side effects with other DMTs. 

The objective of this study was to evaluate the clinical records of patients with RRMS who began treatment with RTX versus other second-line treatments. 

## 2. Materials and Methods

### 2.1. Participants and Procedures

Study Design: Historical cohort study.

Data collection: We included the clinical records of patients of the Neurology Service of the Centro Medico Nacional de Occidente (CMNO) with a diagnosis of RRMS (McDonald Criteria, 2017), who attended the clinic from November 2017 to October 2019. We included the clinical files of patients who started treatment with RTX, NTZ, or FTY for a mean of 12 months uninterruptedly. Patients with incomplete records were excluded. RRMS patients were selected to represent real-world experience in clinical practice in our country. Quality controls were performed through a second review of a subgroup of charts to confirm the accuracy of outliers and the consistency of the data collection. 

Treatment Specifications: At our center, the therapeutic protocol for FTY is 0.5 mg per 24 h orally, while that for NTZ is 20 mg/mL every 28 days intravenously (IV). The clinical protocol establishes that only patients with JC virus levels < 1.5 in serological tests may be included in NTZ treatment. RTX starts with an IV infusion of 1000 mg (as induction) divided into two doses: 500 mg on day 1 and 500 mg on day 14. Afterward, one RTX dose of 1000 mg is administered every 6 months. 

Outcome: The treatment effect was evaluated using the Expanded Disability Status Scale (EDSS), with physicians certified for EDSS assessment (https://www.neurostatus.net/ (accessed on 10 November 2017)), and the annual relapse rate. At our center, magnetic resonance imaging (MRI) with gadolinium is performed as part of the protocol for MS patients every year and a half, so it was impossible to obtain the evaluation to compare the changes caused by RTX. For our study, relapse was defined as any new or worsening neurological symptom that lasted for more than 24 h in the absence of fever or infection [20].

### 2.2. Statistical Analysis

The Statistical Package for Social Sciences SPSS version 25.0, IBM Corp., (Armonk, NY, USA) was used for the analysis. Statistical significance was defined as a *p*-value < 0.05 and is shown along with the 95% confidence interval (CI). The variables are presented as means and standard deviations. Student’s t-test was used to compare the mean delta time, years elapsed with the disease, and initial EDSS at a mean of 12 months between the two groups; the χ^2^ test was used to compare the frequencies; and the Mann–Whitney U test was performed to evaluate the EDSS at baseline and after a mean of 12 months of treatment for each group. Univariate Cox regression analyses were performed, and multivariate model backwards likelihood ratios (LRs) were calculated to determine possible variables related to the risk of relapse in patients with RTX. As covariates, gender, age, EDSS, and the annual relapse rate were considered.

## 3. Results

We reviewed 579 clinical files of patients with MS; 450 of those showed the RRMS phenotype, and 348 patients were prescribed other DMTs during the follow-up time of the cohort and thus were excluded from our study. Our study included 102 patients who started treatment with RTX, NTZ, or FTY due to them experiencing therapeutic failure with other DMTs, more than two relapses in a year, contraindications, or side effects. We classified our patients into two groups: Group 1 included 44 clinical records of patients treated with RTX. Group 2 consisted of 58 clinical records of 10 patients who were treated with NTZ and 48 who were treated with FTY. The sociodemographic and clinical characteristics of the two groups were compared (Table 1). 

The mean age was 39.3 ± 10.6 for patients with RTX and 32.7 ± 8.3 years for patients with NTZ or FTY (*p* < 0.001). The distribution by gender presented differences: women represented 45.5% and 65.5% of the individuals in each group, respectively (*p* = 0.047). The patients receiving RTX had a male:female ratio of 1.2:1, while in the patients of the NTZ or FTY group, this ratio was 1:1.9.

The number of relapses was lower in the group receiving RTX, at 25.0%, while for NTZ or FTY, it was 93.1% of cases (*p* < 0.001). The disease duration from diagnosis to last follow-up in patients with NTZ or FTY was higher (<0.001) than in the RTX group (12.6 ± 4.7 vs. 7.3 ± 6.0, respectively). The comparison of the time of the initial evaluation minus the final one (ΔTime) showed no differences. The ΔTime was 12.7 ± 1.2 months for patients with RTX and 12.5 ± 2.0 for patients with for NTZ or FTY; however, the last file records showed patients who finished the treatment in a maximum of 15 months.

In our study, the RTX group included one patient with levels of JC virus >1.5 who did not wish to continue with NTZ due to the risk of progressive multifocal leukoencephalopathy, thirty-six patients with failure of treatments who were not candidates for NTZ or FTY, and seven naïve patients with EDSS scores of 5.5 (Table 2).

Statistically significant differences were found (*p* < 0.001) when comparing the initial EDSS score between patients given NTZ or FTY and patients given RTX (3.2 ± 1.8 vs. 5.9 ± 1.5, respectively), the final score (4.0 ± 1.7 vs. 5.5 ± 1.6), and the change in EDSS at 12 months after treatment (Figure 1). The ΔEDSS of NTZ or FTY was 0.5 ± 1.5, but we did not find changes in the value of EDSS in patients with RTX after 12 months.

Cox model backwards LR correlation was performed to analyze relapses. Only a correlation between treatment and relapses of the disease was observed (Table 3), with a hazard ratio of 0.284 (95% CI: 0.149–0.545). Other variables such as age, disease duration, and sex were analyzed; however, no correlation was found for them (Figure 2). 

## 4. Discussion

In this retrospective, real-world study, we looked at the effect of RTX on patients treated with RRMS compared to those treated with NTZ or FTY. We observed that RTX had a favorable effect on the EDSS score and annual relapse rate in patients receiving RRMS compared to those receiving NTZ or FTY treatments.

Our results show a statistically significant difference (*p* < 0.001) when comparing the EDSS scores of RTX patients vs. NTZ or FTY patients. The group receiving RTX presented a higher score because the administration of RTX, according to the protocol of our center, was performed only for patients with EDSS scores greater than 5.5 or who presented a more active disease or therapeutic failure with other DMTs.

By comparing the ΔEDSS values at the beginning and after 12 months of treatment between the groups, we identified that, for NTZ or FTY patients, the score on the EDSS score increased (from 3.2 ± 1.8 to 4.0 ± 1.7), while for patients treated with RTX, this score decreased (from 5.9 ± 1.5 to 5.5 ± 1.6), with a statistical significance of *p* < 0.001. Similar studies showed that for patients treated with RTX, the disease does not stop but prevents changes in the patients’ EDSS, reducing the progression of MS [21,22]. 

Statistically significant differences (*p* < 0.001) were found in the annual relapse rate. A decrease of 25.0% vs. 93.1%, respectively, was observed in the annual relapses of the RTX group compared to the NTZ or FTY group. These results were not conclusive because the groups had different clinical characteristics and disease activity. The patients with RTX had five years less with the disease, the sex proportion was different between the groups, and the EDSS score was higher than in patients with NTZ or FTY. The frequency of relapses decreased in male patients, as did the time since diagnosis [20]. 

Some authors observed that patients treated with RTX had fewer relapses compared to those treated with NTZ and FTY. Grandquivist M et al. (2018) performed clinical comparisons of patients treated with RTX vs. NTZ and FTY; although fewer relapses occurred in the patients receiving RTX, the difference was not statistically significant. It should be noted that the induction dose of RTX was variable; some received a single dose of 500 mg or 1000 mg twice a year, which was lower than that applied at our center [19]. Alping P. et al. (2016) analyzed an observational cohort of 256 Swedish patients receiving RTX versus FTY after NTZ; they showed that patients who received RTX had significantly fewer clinical relapses (2% of patients who received RTX had relapses vs. 18% with FTY); furthermore, they observed fewer MRI lesions and adverse effects in patients with RTX, suggesting that RTX may be a valid treatment option [23]. 

According to Cox logistic regression analysis, RTX offers 72% protection against the occurrence of an annual relapse compared to other treatments such as NTZ and FTY. If the patient has fewer relapses, the activity of the disease is lower and, therefore, the progression or high activity of the disease is stopped, thus preventing disability, improving the morbidity and quality of life of patients, and allowing them to become independent in their activities of daily living. However, other factors are involved in the course of the disease, such as the influence of progression independent of relapse activity and other degenerative components, were not explored in this study. Sex, age, and disease duration were also analyzed using Cox logistic regression; however, no relationship was found for these covariates.

Our study shows that the groups treated with RTX and NTZ or FTY were not homogeneous; our results show differences in age, sex, and years with the disease. The main causes of these differences are the characteristics of the patients to whom RTX was administered (EDSS greater than 5.5, therapeutic failure, contraindications, or secondary effects to other DMTs). As this was a real-world study, we could not select patients with the same conditions.

We observed that the patients with RRMS corresponded in age to the economically active population. A statistically significant difference was found (*p* = 0.047) when comparing the male:female proportions in the group treated with RTX (1.9:1) vs. those receiving NTZ or FTY (1:1.2). Patients with RTX showed a higher proportion of women, as MS is more common in females. Additionally, sex is a risk factor for the earlier presentation of the disease, the RRMS phenotype [24], and the frequency of relapses [25]. However, we did not find any correlation between sex and relapses.

The efficacy of RTX worldwide and specifically in Latin America needs to be evaluated, as this treatment is frequently used in clinical practice. Our real-world study is essential because, in Mexico, the few reports on the matter contribute to the challenges of clinical practice. In our country, the treatment of MS is granted based on the treating neurologist’s opinion, the activity of the disease, and the socioeconomic level [26]. Therefore, treatment with RTX has been used as a therapeutic option in our center to treat patients whose options are limited, since the drug is available through federal resources due to its low cost and application in other pathologies.

### Limitations

Although our study suggests that RTX decreases the EDSS score and the incidence of relapses, it also has inherent limitations. This was a retrospective analysis of medical records and there was no control group or randomization, limiting the strength of our conclusions. A randomized, double-blind clinical trial must be performed to assess the efficacy of RTX. We did not register the rate of relapses before the initiation of treatment, so we could not perform a comparison of the effect of treatment on relapses. However, patients who started treatment with RTX, NTZ, or FTY previously presented more than two relapses in a year, which we define as treatment failure. The follow-up period was one year, which favored the detection of short-term benefits but not the effects observed with long-term follow-up. For our study, adherence was not thoroughly examined or adjusted for, which could have affected the results obtained for disease activity. However, the patients receiving RTX and NTZ therapy were promptly summoned for the application of the drugs in the hospital. The method used to verify the treatment adherence of FTY involved a medication count. Before applying the drugs, the potential presence of any new active disease could have influenced the results. Another limitation was the evaluation of MRIs, since they were not obtained due to the routine intervals used; however, we consider such evaluation of utmost importance for evaluating the effect of RTX in RRMS patients. Despite these limitations, the study provides insights into the real-world experience of using RTX in a large group of patients with MS.

## 5. Conclusions

We found a significant clinical improvement during an average of one year of follow-up with RTX treatment, observing a reduction in EDSS and relapses in patients with RRMS with treatment failure or other conditions that limit the administration of line treatments. This observational study is essential in our clinical practice, since patients with limited treatment options represent a challenge in the management of medical care. However, clinical trials and prospective studies with long follow-up periods are necessary to provide sufficient evidence regarding the efficacy of RTX, and thus include this treatment in the therapeutic profile of patients with MS.

## Figures and Tables

**Figure 1 jcm-11-03584-f001:**
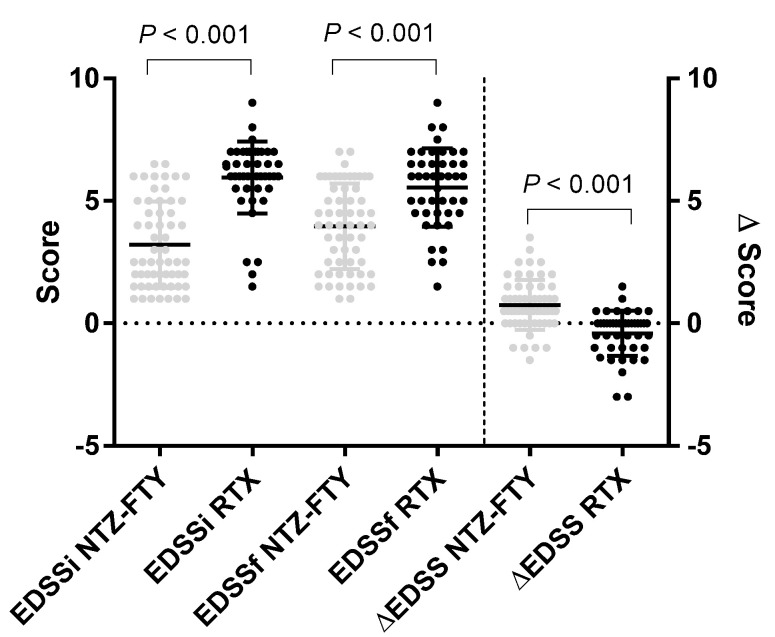
Box plots show medians, quartiles, and 25th and 75th percentiles. EDSSi (initial EDSS); EDSSf (final EDSS); ΔEDSS (change in EDSS after 12 months of treatment; NTZFTY (natalizumab or fingolimod); RTX (rituximab).

**Figure 2 jcm-11-03584-f002:**
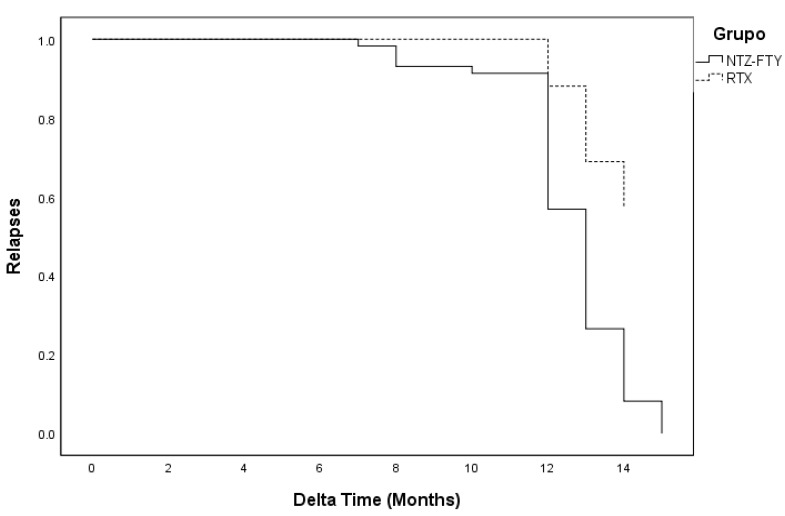
Cox regression model for relapses with ΔTime.

**Table 1 jcm-11-03584-t001:** Sociodemographic and clinical characteristics of MS patients.

Characteristics	Patients with RTX (*n* = 44)	Patients with NTZ of FTY (*n* = 58)	*p*
Age (x ± D.E.)	39.3 ± 10.6	32.7 ± 8.3	0.001
Sex (male/female)	24/20	20/38	0.047
Number of relapses	(11)	(54)	
0	33	4	
1	7	31	<0.001
2	3	16	
3	0	5	
4	1	1	
5	0	1	
ΔTime (months)	12.7 ± 1.2	12.5 ± 2.0	0.598
Years with the disease	7.3 ± 6.0	12.6 ± 4.7	<0.001
EDSS initial	5.9 ± 1.5	3.2 ± 1.8	<0.001
EDSS 12 months	5.5 ± 1.6	4.0 ± 1.7	<0.001

**Table 2 jcm-11-03584-t002:** Treatment before the initiation of RTX.

Treatment	Patients with RTX *n* (%)	Duration of Treatment before RTX (Months) mean ± SD	Patients with NTZ *n* (%)	Duration of Treatment before NTZ (Months) mean ± SD	Patients with FTY *n* (%)	Duration of Treatment before FTY (Months) mean ± SD
NAIVE *	7 (15.9)	-	-	-	19 (39.6)	-
Interferon	10 (22.7)	52.5 ± 44.1	4 (40)	27.6 ± 30.4	21 (43.8)	35.4 ± 31.6
Glatiramer acetate	14 (31.8)	34.6 ± 29.1	3 (30)	37.6 ± 10.9	6 (12.5)	55.8 ± 30.1
Mitoxantrone	3 (6.8)	3.0 ± 2.8	-	-	-	-
Azathioprine	2 (4.6)	38.3 ± 40.5	1 (10)	13.0	2 (4.2)	37.2 ± 35.6
Natalizumab	4 (9.1)	14.8 ± 10.1	-	-	-	-
Fingolimod	4 (9.1)	22.0 ± 4.2	2 (2)	31.8 ± 7.6	-	-

* Naive patients without prior treatment.

**Table 3 jcm-11-03584-t003:** Cox regression analysis.

	β	Standard Error	Hazard Ratio exp (β)	95% CI	
Group 0 = Patients with Rituximab 1 = Patients with NTZ or FTY	−1.257	0.331	0.284	0.149	0.545

Dependent variable—relapses; mode—backwards LR; exclusionary variables—age, disease duration, sex.

## Data Availability

Not applicable.

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
