# Peer review of "Effect of Rituximab Compared with Natalizumab and Fingolimod in Patients with Relapsing–Remitting Multiple Sclerosis: A Cohort Study"

_jcm, 2022, doi:10.3390/jcm11133584_

Round 1

Reviewer 1 Report

Dear authors 

The manuscript "Comparison of the Effectiveness of Rituximab versus Natalizumab and Fingolimod in Patients with Relapsing–Remitting Multiple Sclerosis: A Cohort Study" nicely presents data on clinical effectiveness of RTX compared to NTZ and FTY. However, there are some flaws in the manuscript, which have to be addressed. See in more detail below.

Thank you for your work.

Best regards

Major aspects

  • only patients with negative JCV serology on NTZ were included (line 72). why did you chose only those patients? please elaborate and mention, how many patients you had to exclude initially and during follow up. 
  • you mention that only RRMS patients were selected to be representative of real-world experience (line 78). Please elaborate, how exactly the patients for this study were selected, was there possibly a certain selection bias?
  • one of the weaknesses of the study is certainly, that relapses were defined solely on a clinical basis and that MRI images were not included in the analysis. 
  • another weakness is that the compared patient groups have very different baseline characteristics, e.g. the RTX patients have a much higher EDSS. It is well established, that in more advanced disease annual relapse rates decline.
  • you mention that "if a patient has fewer relapses, the activity of the disease is lower and, therefore, the progression or high activity of the disease is stopped, thus preventing disability" (line 162). You fail to mention the possible influence of the "progression independent of relapse activity" (PIRA) concept on the disease course and also the degenerative components of the disease which are also relevant in relapsing patients.

Minor aspects

  • you write "which can be total or partial". Please rephrase this sentence or elaborate, what you define as a partial neurological disability (line 35)
  • you mention opportunistic infections and other adverse effects related to FTY, but fail to mention the risk of PML, which exists in FTY too (line 47)
  • you mention that RTX shows better clinical efficacy than dimethyl fumarate, but the then cited OLYMPUS trial does not show this finding, please adapt this citation (line 51)
  • what do you mean by disease course years on line 111? Please rephrase.
  • you write "Patients who were treated with RTX had significantly lower EDSS scores" (line 225). But according to your data this is not true, EDSS score was higher, but delta EDSS was smaller. Please rephrase.

Author Response

RESPONSE TO REVIEWERS

We extended our deep thanks to our reviewers. We have attended to all of them gladly and punctually. You will find an in-depth response to each comment, detailing the corresponding modifications made in the text. The other revisor made some other comments, and we modified the text according to both comments.

Major aspects

  • only patients with negative JCV serology on NTZ were included (line 72). why did you choose only those patients? please elaborate and mention, how many patients you had to exclude initially and during follow-up. 

NTZ treatment is associated with the risk of progressive multifocal leukoencephalopathy (PML), an opportunistic infection caused by JCV. Our wording was incorrect, we consider that the JC virus level <1.5 is an adequate value to prevent PML in these patients. It is even one of the protocol requirements of our center to authorize NTZ treatment. We add this information in line 94.

  • you mention that only RRMS patients were selected to be representative of real-world experience (line 78). Please elaborate, on how exactly the patients for this study were selected and was there possibly a certain selection bias?

Our hospital cares for around 579 patients with MS and 450 of those were RRMS phenotype and it is the most representative of our population. At the cohort time of our study, only 102 patients modified treatment with RTX, NTZ or FYT, in the others there was no change in the prescribed medication. We add this information to the document.

  • one of the weaknesses of the study is certainly that relapses were defined solely on a clinical basis and that MRI images were not included in the analysis. 

We explain this weakness in the “Material and methods” and “Discussion” sections. MRI is performed as part of our hospital protocol every 18 months and we followed our patients for only 12 months, so it was not possible to obtain an evaluation to compare changes.

  • another weakness is that the compared patient groups have very different baseline characteristics, e.g. the RTX patients have a much higher EDSS. It is well established, that is more advanced diseases annual relapse rates decline.

According to the protocol of our hospital, RTX is only authorized for MS patients with EDSS > 5.5 or with therapeutic failure to other treatments. Therefore, it was not possible to have similar characteristics in both groups. We explain this in the limitations section.

  • you mention that "if a patient has fewer relapses, the activity of the disease is lower and, therefore, the progression or high activity of the disease is stopped, thus preventing disability" (line 162). You fail to mention the possible influence of the "progression independent of relapse activity" (PIRA) concept on the disease course and also the degenerative components of the disease which are also relevant in relapsing patients.

Kappos in 2018 proposed to evaluate Timed 25-Foot Walk, 9-Hole Peg Test, and EDSS at 30 days and 90 days, and the PIRA concept was established until 2020. Our study included patients from November 2017 to October 2019 and that evaluation was not considered on the clinical records at those times.

Minor aspects

  • you write "which can be total or partial". Please rephrase this sentence or elaborate, on what you define as a partial neurological disability (line 35).

The information was modified according to the bibliography.

  • you mention opportunistic infections and other adverse effects related to FTY but fail to mention the risk of PML, which exists in FTY too (line 47)

We modified the text.

  • you mention that RTX shows better clinical efficacy than dimethyl fumarate, but the then cited OLYMPUS trial does not show this finding, please adapt this citation (line 51)

We apologize for these errors. We revised the bibliography and adapted the text.

  • what do you mean by disease course years on line 111? Please rephrase.

We change “disease course years” to “the disease duration from diagnosis to last follow up”.

  • you write "Patients who were treated with RTX had significantly lower EDSS scores" (line 225). But according to your data this is not true, the EDSS score was higher, but the delta EDSS was smaller. Please rephrase.

 We refer to the delta. The text was modified.

Reviewer 2 Report

The manuscript is about an important topic, examining the effectiveness of rituximab, an off-label, B-cell depleting therapy against natalizumab and fingolimod. While the original idea is important, I have several issues with the presented data and the manuscript itself. First, it should go through a rigorous English language editing.

Second, I present my greatest concerns grouped by the sections of the manuscript.

Introduction:

  1. „…characterized by episodes of reversible neurological disability, which can be total or partial and last from days to weeks”
    What is total and partial disability? I would strongly advise the authors to use the standard definition of the clinical relapse instead of this sentence.
  2. “There are currently more than 11 disease-modifying therapies (DMTs) for preventing the accumulation of CNS lesions and, therefore, transient and/or permanent neurological deficits according to the evolution of the disease”
    I would suggest using the exact number of FDA/EMA approved DMTs, not more than 11. Also, the phrasing is quite odd. These DMTs were proven by (at least) 2 RCTs that they are effective. Effectiveness was proven by measuring the annualized relapse rate, the EDSS progression and the accumulation of new and/or enlarging T2-hyperintense and Gd-enhancing lesions, the classical endpoints of these studies, which these drugs all lowered significantly. I would suggest referring to these results.
  3. “However, none of the currently available therapies completely prevent or reverse neurological impairment.”

Again, an odd phrasing, because that we just don’t really know. For example, in case a patient received a given DMT 10 years ago and since then, shows no measurable disease activity or progression, we may say that the DMT completely prevented further neurological impairment. I would rephrase this sentence, in a similar manner to: “however, there is still no proven curative therapy of the disease”.

  1. “Studies have shown that RTX is effective in the progressive stages of the disease”

This is false. Despite RTX has shown some minor benefits against placebo in some subgroup analysis, the overall OLYMPUS study failed. RTX has no proven efficacy in progressive MS!

  1. “additionally, it has shown better clinical efficacy than first-line DMTs and dimethyl fumarate”

There is no reference provided for this statement. The given reference is the HERMES study, that was a phase II study of RTX against placebo. Please, provide reference for this!

Methods:

  1. The inclusion and exclusion criteria are unclear.
  2. It is unclear, NTZ and FTY patients were included at the initiation of the DMT, or they have been treated with the drug? I suspect the first, yet the phrasing is misleading.
  3. The biggest problem however, is that we have no information on the relapse rate of the groups before the initiation of RTX or NTZ/FTY. It greatly influences the results, yet I find no information on it.

Results and Discussion

  1. It is extremely surprising and honestly, highly doubtful, that during the first year of treatment of the highly effective drugs FTY or NTZ only less than 7% of the patients remain relapse-free! It is completely against all other results in the literature and every reported data from clinical practices (see the cited study in the discussion for one example)! There must be some incredibly serious selection bias or other distorting factor beside the low number of included patients. This should raise questions in the authors about some methodological problems and must be thoroughly documented and assessed, before any further conclusions can be drawn.
  2. There is a discrepancy between the EDSS scores presented in Table 1 and in the text and no explanation for it. Once the initial EDSS of FTY/NTZ patients is 3,2±1,8 in the Table, while it is 2,5±3,5 in the text, for example. Which numbers are the correct one? If there is some subgroup analysis, why it is not mentioned anywhere? These should be clarified.
  3. I see no presented data regarding the Cox-model only the Figure, yet it is presented and discussed in the Discussion section. Where are the calculated LR? What does the Figure represent? And there are some discrepancies again. For example, during a 12-months follow-up, how can the delta time be 20 months? These are issues that have no explanation in the manuscript.
  4. Disease duration, age and sex were not influencing factors for the relapse rate, according to the authors. Despite this, the difference of the sexes and the disease duration is thoroughly discussed in itself, not in clear accordance with the relapse rate. Why is that?
  5. “Patients treated with RTX had disease courses of 7.3±6.0 years, which is less (<0.001) than that for patients treated with NTZ or FTY; this means that the patients treated with RTX with the RRMS phenotype had more aggressive disease, so it was necessary to start with a more advanced treatment.” RTX is not even approved in MS, how could it be considered a more advanced treatment than NTZ?

To conclude the review, the manuscript – while raising an important topic, the effectiveness of the off-label, yet quite often used drug, rituximab against two highly effective DMTs for MS – is confused, the methods unclear, the presentation of the results is lacking on one hand while on the other, raises the possibility of some serious distorting issues or selection bias which should be thoroughly addressed before publication.

Author Response

RESPONSE TO REVIEWERS

We extended our deep thanks to our reviewers. We have attended to all of them gladly and punctually. You will find an in-depth response to each comment, detailing the corresponding modifications made in the text. The other revisor made some other comments, and we modified the text according to both comments.

Dear revisor 2:

We extended our deep thanks to our reviewers. We have attended to all of them gladly and punctually. You will find an in-depth response to each comment, detailing the corresponding modifications made in the text. The other revisor made some other comments, and we modified the text according to both comments.

Introduction:

  1. „…characterized by episodes of reversible neurological disability, which can be total or partial and last from days to weeks” What are total and partial disability? I would strongly advise the authors to use the standard definition of clinical relapse instead of this sentence.

The information was modified according to the bibliography.

  1. “There are currently more than 11 disease-modifying therapies (DMTs) for preventing the accumulation of CNS lesions and, therefore, transient and/or permanent neurological deficits according to the evolution of the disease” I would suggest using the exact number of FDA/EMA approved DMTs, not more than 11. Also, the phrasing is quite odd. These DMTs were proven by (at least) 2 RCTs that they are effective. Effectiveness was proven by measuring the annualized relapse rate, the EDSS progression, and the accumulation of new and/or enlarging T2-hyperintense and Gd-enhancing lesions, the classical endpoints of these studies, which these drugs all lowered significantly. I would suggest referring to these results.

We described the NTZ and FTY efficacy and we added references. We update the number of disease-modifying therapies approved by FDA/EMA; according to the references found.

  1. “However, none of the currently available therapies completely prevent or reverse neurological impairment.” Again, this an odd phrasing, because we just don’t know. For example, in case a patient received a given DMT 10 years ago and since then, shows no measurable disease activity or progression, we may say that the DMT completely prevented further neurological impairment. I would rephrase this sentence, in a similar manner to: “however, there is still no proven curative therapy of the disease”.

We made the suggested changes.  

  1. “Studies have shown that RTX is effective in the progressive stages of the disease”. This is false. Despite RTX has shown some minor benefits against placebo in some subgroup analysis, the overall OLYMPUS study failed. RTX has no proven efficacy in progressive MS!

“additionally, it has shown better clinical efficacy than first-line DMTs and dimethyl fumarate” There is no reference provided for this statement. The given reference is the HERMES study, that was a phase II study of RTX against placebo. Please, provide reference for this!

We added the correct bibliography and we modified the text.

Methods:

  1. The inclusion and exclusion criteria are unclear.

We added the Data collection section.

  1. It is unclear, NTZ and FTY patients were included at the initiation of the DMT, or if they have been treated with the drug? I suspect the first, yet the phrasing is misleading.

We added the Treatment Specifications and data collection section.

  1. The biggest problem, however, is that we have no information on the relapse rate of the groups before the initiation of RTX or NTZ/FTY. It greatly influences the results, yet I find no information on it.

We apologize for that, but only we register that the patients had at least two relapses more two relapses in a year.

Results and Discussion

  1. It is extremely surprising and honestly, highly doubtful, that during the first year of treatment of the highly effective drugs FTY or NTZ only less than 7% of the patients remain relapse-free! It is completely against all other results in the literature and every reported data from clinical practices (see the cited study in the discussion for one example)! There must be some incredibly serious selection bias or other distorting factor beside the low number of included patients. This should raise questions in the authors about some methodological problems and must be thoroughly documented and assessed, before any further conclusions can be drawn.

We added the “Limitations” section and added other research about relapses.

  1. There is a discrepancy between the EDSS scores presented in Table 1 and the text and no explanation for it. Once the initial EDSS of FTY/NTZ patients is 3,2±1,8 in the Table, while it is 2,5±3,5 in the text, for example. Which numbers are the correct one? If there is some subgroup analysis, why it is not mentioned anywhere? These should be clarified.

We apologize that we exchanged the data.

  1. I see no presented data regarding the Cox model only the Figure, yet it is presented and discussed in the Discussion section. Where are the calculated LR? What does the Figure represent? And there are some discrepancies again. For example, during a 12-months follow-up, how can the delta time be 20 months? These are issues that have no explanation in the manuscript.

We added to table 3 and we had a follow-up for 15 months. We adjust the scale of the graph. This is because the application of treatment was delayed in some patients.

  1. Disease duration, age and sex were not influencing factors for the relapse rate, according to the authors. Despite this, the difference of the sexes and the disease duration is thoroughly discussed in itself, not in clear accordance with the relapse rate. Why is that?

We made the suggested changes. 

  1. “Patients treated with RTX had disease courses of 7.3±6.0 years, which is less (<0.001) than that for patients treated with NTZ or FTY; this means that the patients treated with RTX with the RRMS phenotype had more aggressive disease, so it was necessary to start with a more advanced treatment.” RTX is not even approved in MS, how could it be considered a more advanced treatment than NTZ?

We modified the text.

To conclude the review, the manuscript – while raising an important topic, the effectiveness of the off-label, yet quite often used drug, rituximab against two highly effective DMTs for MS – is confused, the methods unclear, the presentation of the results is lacking on one hand while on the other, raises the possibility of some serious distorting issues or selection bias which should be thoroughly addressed before publication.

Round 2

Reviewer 1 Report

The authors adressed all the major and minor aspects, therefore I suggest to accept the manuscript in this form.

Author Response

We appreciate the time and effort you and the reviewers put into providing feedback on our manuscript to improve it.